# Scoping Review on Barriers and Challenges to Pediatric Immunization Uptake among Migrants: Health Inequalities in Italy, 2003 to Mid-2023

**DOI:** 10.3390/vaccines11091417

**Published:** 2023-08-25

**Authors:** Samina Sana, Elisa Fabbro, Andrea Zovi, Antonio Vitiello, Toluwani Ola-Ajayi, Ziad Zahoui, Bukola Salami, Michela Sabbatucci

**Affiliations:** 1School of Public Health, University of Alberta, 11405 87 Ave NW, Edmonton, AB T6G 1C9, Canada; ssana@ualberta.ca (S.S.); olaajayi@ualberta.ca (T.O.-A.); zahoui@ualberta.ca (Z.Z.); bukola.salami@ualberta.ca (B.S.); 2Internationalization Staff Unit, Institutional Services Area, University of Trieste, Piazzale Europa, 1, 34127 Trieste, Italy; 3Area Science Park, Padriciano, 99, 34149 Trieste, Italy; 4Directorate General for Hygiene, Food Safety and Nutrition, Ministry of Health, Viale Giorgio Ribotta 5, 00144 Rome, Italy; zovi.andrea@gmail.com (A.Z.); 5Directorate General for Health Prevention, Ministry of Health, Viale Giorgio Ribotta 5, 00144 Rome, Italy; 6Department of Community Health Sciences, Cumming School of Medicine, University of Calgary, 3280 Hospital Drive, Calgary, AB T6G 1C9, Canada; 7Department Infectious Diseases, National Institute of Health, 00160 Rome, Italy

**Keywords:** migrant children, vaccination equity, health strategies, immunization, vaccine-preventable diseases (VPD), scoping review, Italy

## Abstract

In the aftermath of the COVID-19 pandemic, asylum seekers, refugees, and foreign-born migrants are more likely to suffer from physical, mental, and socioeconomic consequences owing to their existing vulnerabilities and worsening conditions in refugee camps around the world. In this scenario, the education of migrants and newcomers about immunization is critical to achieving health equity worldwide. Globally, it is unclear whether government vaccination policies are prioritizing the health information needs of migrants. We searched for studies investigating the vaccination uptake of migrant children settled in Italy that were published between January 2003 and 25 June 2023. Following Arksey and O’Malley’s five-stage method for scoping reviews, all potentially relevant literature published in English was retrieved from SciSearch, Medline, and Embase. This search resulted in 88 research articles, 25 of which met our inclusion criteria. Our findings indicate unequal access to vaccination due to a lack of available information in the native language of the immigrants’ country of origin, vaccine safety concerns or lack of awareness, logistical difficulties, and fear of legal consequences. The findings strongly encourage further government and political discourse to ensure migrants have fair, equitable, ethical, and timely access to essential medicines.

## 1. Introduction

Monitoring inequalities in health remains central after the COVID-19 pandemic and the continuing emergence of new health crises around the world. Globally, around 1.3 million children died in 2017 from vaccine-preventable diseases (VPDs). Indeed, one in five children worldwide do not have access to immunization, which is widely considered a beneficial, low-cost investment [1]. Despite this, the implementation of immunization programs varies greatly across countries and communities, contributing to health inequalities. The World Health Organization (WHO) estimates the COVID-19 pandemic has had a serious impact on immunization activities [2]. The most recent report released by UNICEF indicates that 67 million children worldwide missed either some or all routine vaccinations between 2019 and 2021, with 48 million children not receiving a single dose during this time period [3]. The children left unvaccinated are often those who are the most at risk: the poorest, the most marginalized, and those affected by conflict or forced migration. More specifically, the demographic most at risk of severe health consequences due to under-immunization or no immunization is immigrant and migrant children, likely from countries with destabilized or poorly functioning healthcare systems. Among newcomer populations, the proper vaccine uptake and immunization of school-aged children against infectious diseases is a growing area of interest in international preventative health policies in the last decade [4,5,6,7,8]. Indeed, health inequality causes effects that accumulate over the life course and can transfer across generations. A life-course approach increases the effectiveness of interventions, bringing returns for public health and the economy.

Italy has experienced slight increases in the number of foreign-born migrants since 2015. In 2019, Italy was the fifth-most popular migrant destination in Europe, with over 6 million international migrants accounting for 10% of the total Italian population [9]. These newcomers mostly comprise temporary or permanent immigrants; asylum seekers and refugees, defined as those ‘‘outside his/her country of origin for reasons of feared persecution, conflict, generalized violence, and require international protection” (United Nations definitions https://refugeesmigrants.un.org/definitions, accessed on 10 July 2023); and undocumented migrants who would not be captured in official statistics arrive in Italy every year. In 2018, over 23,000 irregular maritime migrants landed in Italy. Of them, 15% (estimated from January to November 2018) were unaccompanied children. Italy also recorded over 3000 new disaster displacements in 2018 [10].

Often, when newcomers arrive in a new country, they experience difficulties accessing primary and specialized healthcare [11]. Socioeconomic and health conditions of immigrants in Italy have been studied locally. Some studies show that immigrants have poorer health outcomes and higher prevalence of disease than Italy-born individuals. In particular, immigrant children are suffering a high level of psychological distress, exposing them to an increased risk of mental and physical health problems. Despite the recent success in improving national childhood vaccination coverage [12,13], possible inequalities in newcomer childhood immunization remain unclear. To determine progress in childhood immunization benefits in all population groups in Italy, we reviewed all relevant scientific literature published in the last 20 years regarding migrant childhood immunization. Scoping reviews are well suited for areas characterized by limited knowledge or research. We conducted a scoping review of research studies performed in Italy with the aim of providing evidence on strategies and policies addressing newcomer childhood vaccination inequalities and identify favorable and critical factors that can influence decision-making towards improved health equity in Italy. We focused on pediatric vaccinations administered to migrant children who arrived in Italy in the last 20 years.

## 2. Materials and Methods

We conducted this scoping review conforming to the standards outlined in the Preferred Reporting Items for Systematic Reviews and Meta-Analyses (PRISMA) and the Prisma-ScR reporting guidelines found in the PRISMA Extension for Scoping Reviews (PRISMA-ScR): Checklist and Explanation [14], according to Arksey and O’Malley’s stepwise scoping review methodological framework [14,15,16]. We chose a scoping review methodology to collate evidence that fits pre-specified eligibility criteria to answer our research question with the aim to minimize bias and make the available evidence accessible to decision makers in Italy.

We followed five stages in undertaking the scoping review: (1) identification of the research question, (2) collection of relevant studies, (3) selection of studies, (4) charting of data, and (5) development of results: collating, summarizing, and reporting final data. Thematic analysis was used to summarize the results of qualitative studies, using mind-mapping and manual coding (thematic sorting) of the data into useful categories for synthesis. Numerical summary and descriptive statistics were used to summarize the results of quantitative studies. Data from the articles were compared across the coding segments to identify relevant linkages and patterns.

### 2.1. Search Strategy

We searched the online research databases of SciSearch, Medline, and Embase, limiting the search to the 20-year period from 1 January 2003 to 25 June 2023. We used search terms that represent children/pediatric vaccination strategies offered to newcomer children in Italy, as follows:

#### 2.1.1. To Identify Children 0–18 Y of Age

child or toddler or adolescen? or teenager or teens or pediatric or paediatric or baby or newborn or infant or child + nt/ct or minors/ct or adolescent/ct

#### 2.1.2. To Identify Vaccination

vaccine or vaccination or vaccination + nt/ct or vaccines + nt/ct

#### 2.1.3. To Identify Newcomers

emigrant or imigrant or immigrant or refugee or asylum(w)seeker or migrant or newcomer or displaced(w)person or transient(w)person or squatter or nomad or transients and migrants/ct or emigrants and immigrants + nt/ct or emigration immigration + nt/ct

#### 2.1.4. To Identify Italy

Italy or Italian or Sicily or Lampedusa or Messina or Apulia or Lombardy or Milan or Rome or Italy + nt/ct

The above four groups of search terms were combined. Terms were searched with linguistic variants and plural forms. The search was restricted to articles published in English.

### 2.2. Eligibility Criteria

#### 2.2.1. Inclusion Criteria

In the review, we included scientific articles providing information on vaccination policies, strategies, and practices offered to newcomer children aged 0–18 years in Italy at the local/national level.

#### 2.2.2. Exclusion Criteria

We excluded from the review scientific articles providing information on individuals over 18 years of age, or targeting a mixed population of children and adults, or focusing on Italy and other countries. We also excluded conference proceedings, conference abstracts, posters, and correspondence.

### 2.3. Study Selection

Overall, 88 articles were identified based on the keywords selected. Duplicates (41 documents) were automatically removed at the end of the search strategy process. For study screening and selection, a two-step process was followed: the first step was screening the search results by title, abstract, and keywords; the second step involved skimming the full text in consideration of the inclusion criteria. A total of 37 articles were identified after the first step, with this number further reduced to 25 after the second step. Data from these 25 articles are included in the extraction table (Appendix A).

### 2.4. Data Charting Process

Content from the 25 articles was charted in Google Sheet utilizing the following 19 study characteristics: (1) article title, (2) first author name, (3) year of publication, (4) journal name, (5) research question and/or objective, (6) theoretical framework (qualitative or quantitative), (7) methodology (ethnology, quasi-experimental, etc.), (8) method (interviews, focus groups, observations, survey, etc.), (9) sampling (purpose, maximum variation, convenience, random, cluster, etc.), (10) sample size, (11) age of participants, (12) data source (i.e., parent vs. child vs. health professional), (13) clinical area of focus, (14) period of data collected, (15) country of origin or region, (16) destination country or region, (17) summary of findings, (18) summary of implications, and (19) key conclusions. For study characteristics/categories where information was not available, we noted “N/A”.

## 3. Results

### 3.1. Search Results

Overall, we identified 88 studies from the literature search, 63 of which were excluded as they did not meet our inclusion criteria; this left 25 studies to be included in the review (Figure 1).

The number of the relevant studies published and included in our 20-year assessment was highest in the years 2017 to 2019 (Figure 2).

Just under half (N = 12) of the articles were published in one of three journals: *Vaccine* (N = 7), the *International Journal of Environmental Research and Public Health* (N = 3), and *Human Vaccines & Immunotherapeutics* (N = 2). The remaining 13 articles appeared in 13 different journals.

### 3.2. Characteristics of the Included Articles/Studies

Seventeen (68%) of the twenty-five studies were classified as quantitative and eight (32%) as qualitative (Appendix A). Data were collected from questionnaires administered to healthcare professionals in nine studies (36%), by comparing two or more groups of patients in six studies (24%), from information acquired at admission or discharge or screening in five studies (20%), and by using databases in five studies (20%).

Overall, the studies reflect a great deal of heterogeneity with respect to the diseases considered. In most cases, the study objective was to investigate vaccination trends among migrants and analyze the causes that prevent them from accessing full vaccination against major infectious diseases and/or to investigate the type of information recorded when migrants enter the host country. Below, we summarize the main information collected in the studies analyzed: vaccination trends among migrants, causes of low vaccination rates in migrants, the need to implement registration of migrant immunization data, the comparison of health status between regular and irregular migrants, and data concerning the access by migrants to vaccination against hepatitis B (HBV), human papillomavirus (HPV), and other viruses.

### 3.3. Vaccination Trends among Migrants

Vita et al. [17] reported an increase in the number of migrants vaccinated over time, which they attributed to a change in the strategy for administering vaccines in Italy, i.e., upon arrival at the immigrant center rather than by territorial vaccination services at a later stage. The result was a notable impact on vaccination coverage levels. Stampi et al. [18] investigated the vaccination coverage achieved for mandatory and recommended vaccinations in European countries to identify the percentage of unvaccinated individuals. Their analysis shows compulsory vaccinations are administered to migrants without vaccination protection, although they highlight the need to improve the continuation and administration of non-compulsory or multi-dose vaccinations. According to Ercoli et al. [19], the children of immigrants are a large and growing population that lack access to healthcare. Children with illiterate parents are more likely to remain unvaccinated, with about 66% of the children included in this study never having received any kind of healthcare. Hargreaves et al. [20] investigated the state of the process by which vaccinations are administered and the resulting healthcare. Their research shows that access to vaccinations is better for children than adults, and therefore immigrant children are more likely to be protected from VPDs than their parents. Although this is a step in the right direction, the authors conclude that the recovery of vaccination rates for adults is crucial. Fabiani et al. [21] investigated the vaccination coverage rate in immigrant mother–child pairs, which is lower than in Italian mother–child pairs. They analyzed the causes, finding only some observed differences are explained by socio-geographical characteristics. Other factors that may play a role include language barriers, the level of healthcare available during pregnancy, and the ability to contact migrants as they may frequently change their domicile.

### 3.4. Causes of Low Vaccination Rates in Migrants

Mipatrini et al. [22] highlighted the main reasons why migrants and refugees have low immunization rates in Italy compared to European-born individuals. They note low vaccination coverage in the country of origin and several critical issues may limit migrants’ access to vaccination in Europe. Specifically, migrants often move around the continent, posing challenges with respect to receiving the multiple doses at regular times required for many vaccines; the immunization status of migrants can be unknown; often, migrants refuse to be registered at the medical center for fear of legal consequences; a lack of coordination among public health authorities of neighboring countries can often lead to missed vaccination; and sometimes the migrants face severe health and/or economic crises in the hosting country. After describing reasons for migrants’ low immunization rates, the authors focused on these critical issues to be addressed to overcome the disparity. In Giambi et al. [23], the parents of immigrant children were divided into groups according to their views on vaccinations, with the aim of analyzing the underlying causes of the lack of vaccination for some infectious diseases. The authors report doubts about the safety of vaccination as the main reason that prevents parents from having their children vaccinated.

### 3.5. The Need to Implement Registration of Migrants’ Immunization Data

Giambi et al. [24,25] highlighted the lack of vaccination records for immigrants, asylum seekers, and refugees in many countries, and emphasized the need to increase the collection of immunization records for immigrants. Dalla Zuanna et al. [26] investigated the respective differences in the administration of vaccination at the community and local levels. In many centers, a wider range of vaccines than those offered by the national policy were administered at the community level to both children and adults. The main issue concerned the tracking and storage of records of vaccinations administered, even when immigrants move across European countries. Giambi et al. [24] examined immunization policies and practices directed to irregular migrants, refugees, and asylum seekers across the 30 European Union/European Economic Area (EU/EEA) countries. Their findings show information on vaccines administered to migrants is recorded by a wide variety of methods in the different countries; moreover, individual and aggregated data were not shared with other centers/institutions in 13 and 15 countries, respectively. Twenty countries, including Italy, reported they did not collect data on vaccination uptake among migrants, and three countries only had these data at the national level. Thirteen countries only had procedures to ensure migrants’ access to vaccinations at the community level. The study concluded the development of migrant-friendly strategies may facilitate vaccination access as well as data collection, with both approaches necessary to address existing gaps. Ravensbergen et al. [27] focused on the need to adopt harmonized immunization policies between migrants and European citizens. These authors note that disseminating migrant-specific guidance to frontline health workers would need greater emphasis to improve vaccine uptake in migrant populations across the EU/EEA countries.

### 3.6. Comparison of Health Status between Regular and Irregular Migrants

Mipatrini et al. [28] compared access to healthcare for migrants with and without regular documents. Their research shows migrants without regular documents are at a higher risk of being hospitalized for avoidable conditions, including VPDs. Therefore, the authors stress the need to implement programs specifically designed to help migrants without regular documents.

### 3.7. Data concerning Migrant Access to Vaccination against HBV, HPV, and Other Viruses

Napolitano et al. [29] investigated the level of knowledge immigrants in Italy have about HPV vaccination. Their findings show a lack of awareness of the importance of this vaccination can lead to low awareness about the risk of infection among the immigrant community, lack of prevention, and increased risk of infectious disease. Although immigrants had less knowledge of vaccinations than the general public, they were still willing to get the vaccine for themselves and their children if given the opportunity. However, many did not have the same access to the HPV vaccine as other citizens. Gabutti et al. [30] evaluated information about the incidence of pediatric HBV and HPV infections in native-born, immigrant, and adopted children living in Italy. Their analysis shows the level of HBV infection decreased in Italian children over time due to the mandatory vaccination policy. However, this decrease did not occur for immigrant and adopted children due to different strategies in their countries of origin. The authors highlight the importance of spreading socialization and medical information among foreign populations to increase access to vaccinations. Coppala et al. [31] evaluated the incidence of HBV infection in immigrants, showing that some carried the hepatitis B surface antigen (HBsAg) and that some were unaware of their status in this regard. These authors suggest screening should be performed on all immigrants entering Italy to ensure they receive proper healthcare and treatment if found to be carriers of the virus. This approach would require consideration of language barriers, socioeconomic class effects, and cultural barriers. Fabiani et al. [32] investigated rubella vaccination in Italian and immigrant women, finding immigrant women were more likely to be unaware of their immunization status than Italian women. Italian women were immunized against rubella at a higher rate than immigrant women, but socioeconomic factors did not play a role in the observed differences. Access to vaccine screening is free in Italy, so other factors, such as lack of knowledge and accessible information, may be responsible for the difference in rubella vaccination rates.

Myran et al. [33] evaluated the effectiveness of chronic hepatitis B (CHB) screening and vaccination programs among migrants in the EU/EEA. The studies identified through this systematic review indicated the vaccination schedules employed were highly effective in reducing the prevalence of CHB in the vaccinated children. Several studies highlighted the barriers and the strategic gaps to ensure treatment and care for migrants with CHB. Often, migrants lack access to primary healthcare, they can face additional barriers to assistance, and developing screening approaches remains a challenge.

## 4. Discussion

The number of the studies per year included in our 20-year assessment was highest between 2017 and 2019, showing a recent increasing trend in the focus on pediatric vaccinations in migrants. Then, as recognized by the WHO, the COVID-19 pandemic diverted attention from routine pediatric vaccinations worldwide, even among the most fragile populations, including asylum seekers and migrants. In the pandemic period, the number of the relevant studies returned to a baseline of around one publication per year. However, in Italy, the health of migrant populations can have a considerable impact. The number of migrants arriving by sea in Italy was around 20,000 in 2021, over 27,000 in 2022, and over 61,000 as of June 2023. Among them, unaccompanied foreign minors numbered in the thousands every year (over 10,000 in 2021, over 14,000 in 2022, and almost 7000 as of June 2023). In the first half of 2023, 45% of the immigrants to Italy came from Ivory Coast, Egypt, Guinea, Pakistan, and Bangladesh [34]. Overall, our review outlines several issues that need to be addressed to overcome the primary challenges preventing the vaccination of these individuals in Italy: the lack of available information in the native language of immigrants’ country of origin, vaccine safety concerns, the lack of awareness of VPDs, logistical difficulties in accessing vaccination, and the absence of specific centers to administer vaccines to incoming migrants. In particular, children with illiterate parents are more likely to remain unvaccinated. Studies assessing the vaccination rates among migrants show key aspects for establishing effective prevention policies to ensure infection protection and limit outbreaks in host countries. Developing culturally and linguistically congruent health information is critical to raise awareness of children’s vaccination status. One key aspect is to accurately determine the vaccination coverage level of migrants upon their arrival at specific reception centers through information recorded by dedicated healthcare professionals. The responsible use of digital technologies, including artificial intelligence (AI), could support asylum and migration management, including health data registration and availability at the national level [35]. This information can enable subsequent vaccination of migrants by local vaccination services, creating a strong collaboration between asylum seeker centers and local healthcare services.

The studies in this review also demonstrate that, among migrants, access to vaccinations is generally better for children than adults. As a result, an effective vaccination strategy should also consider the need to restore vaccinations for adults with the aim of achieving the highest possible vaccination coverage. To this end, the implementation of effective guidelines for immigrant vaccination in reception centers, which direct migrants to vaccination centers within local healthcare services, is necessary to ensure proper care for immigrants that is on par with all citizens who have access to public healthcare. Furthermore, the development of more technologically advanced electronic records is extremely important and necessary to accurately document vaccinations prescribed by the National Immunization Plan and administered to immigrants. However, data on immunized immigrants are often stored locally in different ways and are not available at the national level. Therefore, further developments are needed to establish adequate data exchange mechanisms to ensure comprehensive vaccination offerings and unnecessary revaccination. There is also a clear need to implement migrant parental access to specific reception centers with effective communication by dedicated expert operators, supporting the services the Italian national health system can provide. Specifically, the limited access to healthcare services by irregular immigrants exposes them to greater risks compared to regular migrants, resulting in inequalities due to the lack of access to the national healthcare system. The need for targeted programs to assist undocumented immigrants arises as an additional crucial requirement to ensure adequate healthcare assistance for this demographic.

In addition, Giambi et al. [24] highlight that, although diverse, strategies for vaccinating migrants are generally being implemented throughout most of the EU/EEA in line with international recommendations. To fill existing gaps and ensure effective information exchange and data sharing among member states, initiatives should be implemented to facilitate sharing information on the number and series of vaccinations administered in the country of origin. The main strategies identified to overcome these problems include the development of digital vaccination registries that can track vaccination trends among migrants, as well as communication campaigns to promote and encourage collaboration among public health authorities in the EU/EEA-based collection of digital data is becoming increasingly relevant given the emergence of new technologies and sources of data, such as social media, mobile phone data, satellite imagery, and AI, as well as the growing need for reliable statistics, comprehensive and timely data, and national preparedness and response plans against infectious threats [35]. However, the creation of national digital databases to collect vaccine coverage information on migrant populations has so far faced many challenges in low- and middle-income countries (LMICs) and even in high-income countries (HICs). For example, the confidentiality of personal data is especially important in sensitive contexts such as child migration, so data protection and privacy policies must be a priority; data protection is an evolving area of law that may have diverse interpretations in various cultures and countries even if in agreement with the Universal Declaration of Human Rights [36] and the Charter of Fundamental Rights and the Treaty on the Functioning of the European Union [37]; harnessing the power of digital data, which must proceed in parallel with modifying existing protection rules or creating complementary legal instruments, making periodic international adjustment impractical; existing legal instruments intended to protect privacy and security of offline data subjects may be considered ineffective, given that borders and boundaries are unclear in the digital space; migrants often move between countries or continents and personal health information may not be identified in every area; many vaccines require multiple doses that make collecting individual data on a complete vaccination cycle even harder; and migrating digital data between platforms or countries increases exposure to several risks, such as data corruption, loss, duplication, or inconsistencies. Clearly, multiple considerations must be made related to the collection of digital data on vaccination coverage in migrant children.

Finally, we note our study is limited by restricting our literary search strategy to publications in English. Our choice was motivated by our desire to focus on studies that had undergone a peer-review process to ensure the quality of the results described.

## 5. Conclusions

Access to medicines, including vaccines, is inherently connected to principles of equality, non-discrimination, transparency, participation, and accountability, as well as availability, efficacy, and safety. Important considerations include addressing newcomer health needs through dedicated centers and healthcare workers as well as reducing vaccine hesitancy through communication and the reinforcement of data from qualified long-term and not-for-profit scientific research. Obstacles related to communication and information, including low digital literacy or lack of technology, language difficulties, poor written/reading literacy, poor doctor–patient communication, lack of interpreter services, and providing information in an accessible and acceptable format, need to be eliminated through information campaigns based on local data and conducted by trained professionals. Overcoming barriers including remoteness, unawareness, parental work commitments, and fear of arrest is also important, as is providing education to parents and training to health staff on the benefits as well as possible side effects of active immunization. Finally, the sharing of electronic records can ensure the accurate registration of vaccinations administered to immigrants, thus avoiding any unnecessary healthcare procedures, including re-vaccination.

## Figures and Tables

**Figure 1 vaccines-11-01417-f001:**
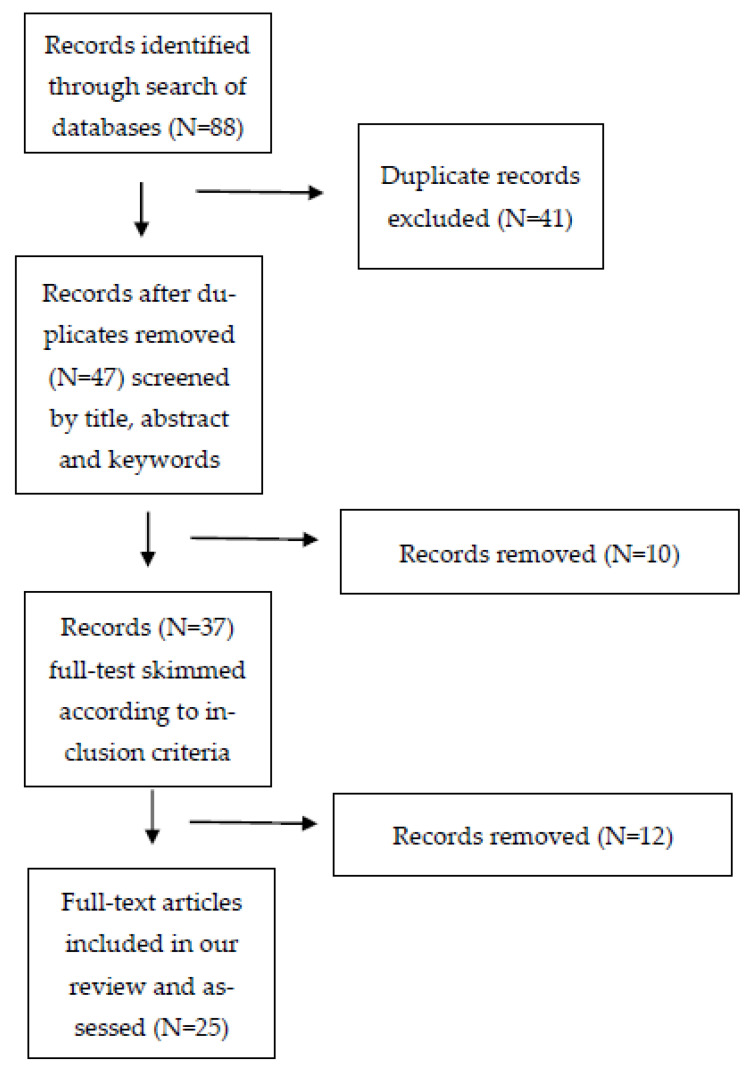
Flow diagram of article selection process for the scoping review on barriers and challenges to pediatric immunization uptake among migrants in Italy, 2003 to mid-2023.

**Figure 2 vaccines-11-01417-f002:**
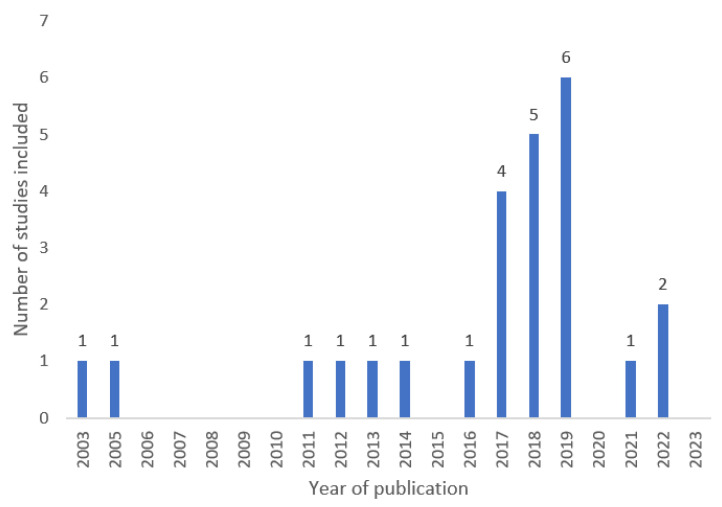
Number of studies published per year included in the review regarding barriers and challenges to pediatric immunization uptake among migrants in Italy, 2003 to mid-2023 (N = 25).

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
