# Peer review of "Scoping Review on Barriers and Challenges to Pediatric Immunization Uptake among Migrants: Health Inequalities in Italy, 2003 to Mid-2023"

_vaccines, 2023, doi:10.3390/vaccines11091417_

Round 1
Reviewer 1 Report
The article deals with an important subject. Methods employed are adequate and results are relevant for public polities.
Regarding conclusions, in my view, the first 2 sentences would be better at the introduction, as they are not conclusions from the scoping review. Additionally, conclusions of the study could be better highlighted.
In line 336 - the word .particularly, for starting a sentence needs some editing.
Author Response
The authors thank the reviewer for the revision. Here below there are a point-by-point response to the comments:
Point 1: Regarding conclusions, in my view, the first 2 sentences would be better at the introduction, as they are not conclusions from the scoping review. Additionally, conclusions of the study could be better highlighted.
Response 1: The authors thank the reviewer for this suggestion. We revised the conclusions as you suggested. Please consider lines 372-388 (highlighted in green).
Point 2: In line 336 - the word particularly, for starting a sentence needs some editing.
Response 1: The authors thank the reviewer for this suggestion. We revised the sentences. Please consider lines 358-367 (highlighted in green).
Besides, we changed the sections of the manuscript that the Editor highlighted in yellow and we extensively reviewed the English language.
Reviewer 2 Report
Good contribution. The main feature of the paper is to highlight some disparities in the health of migrants in a developed country.
The manuscript’s Strengths: Data analyzed based on other scholarly papers, Weaknesses: There are no new findings from the authors.
To improve the manuscript, there are some advice:
1. Can be defined as a research problem more clearly
2. Should be developed a clean analytical framework
3. More awareness needs to be present in the presentation of research findings so that a reader can easily understand what is truly the essence.
I believe that many weaknesses will be removed if the three points above are carefully observed.
Author Response
The authors thank the reviewer for the revision. Here below there are a point-by-point response to the comments:
Point 1: Can be defined as a research problem more clearly
Response 1: The authors thank the reviewer for this suggestion. We revised the sentence. Please consider lines 79-84 (highlighted in green).
Point 2: Should be developed a clean analytical framework
Response 2: The authors thank the reviewer for this suggestion. We grouped the key findings and main criticalities in line with the objectives of our work, i.e., highlighting gaps that can be addressed by policy makers and experts in this field to increase vaccination access and coverage among migrant children in Italy. We have now redrafted the paragraph “Characteristics of the included articles/studies” with a clear analytical framework. Please consider lines 170-185 (highlighted in green).
Point 3: More awareness needs to be present in the presentation of research findings so that a reader can easily understand what the essence is truly.
Response 3: The authors thank the reviewer for this comment. However, they do not understand what the reviewer is asking in terms of revision/amendment. The authors note the results section has already been divided into subsections according to the main findings extracted from the studies, i.e., they have been grouped according to the topic covered and the key findings highlighted. It was not deemed necessary to list other studies in the results section, as an attempt was made to provide the reader with a comprehensive but concise overview of the most relevant findings from the selected studies. If, in the reviewer's opinion, other studies with relevant findings should be highlighted in the manuscript (see supplementary file listing all studies), please notify the authors so they can promptly incorporate them into the article.
Besides, we changed the sections of the manuscript that the Editor highlighted in yellow, and we extensively reviewed the English language.
Reviewer 3 Report
This paper is a systematic review of the low vaccination coverage of immigrant children in Italy.
While it does a good job of summarizing the evidence already reported, there are two problems.
First, although it is in the form of a systematic review, the paper deals only with surveys and does not integrate the results, leaving it as a simple review. The discussion is also limited to a list of vaccination issues of immigrant children. The study was limited to studies conducted in Italy, which raises the question of whether a systematic review method was necessary for the study design? This point needs to be clarified in the method.
Second, the international sharing of digital information on vaccination histories of those from other countries would certainly be a decisive solution. But it is obvious that even if it were an ideal idea, it would be extremely difficult to achieve practically. Even in high-income countries, only a few countries have centralized management of digital information. It is too much to expect lower middle income countries and territories, where even the management of family registers is inadequate, to manage individual vaccination histories using digital information and then share it with the rest of the world. It is necessary not only to talk about a dream, but also to clearly explain the practical challenges to realize the dream in the discussion.
Author Response
The authors thank the reviewer for the revision. Here below there are a point-by-point response to the comments:
Point 1: Although it is in the form of a systematic review, the paper deals only with surveys and does not integrate the results, leaving it as a simple review.
Response 1: The authors thank the reviewer for this comment. As recognized by the reviewer, we conducted the search with all the rigorous steps of a systematic review. Results from the eligible studies meeting the inclusion criteria were analyzed as summarized in lines 170-175 (highlighted in green): “Data were collected from questionnaires administered to health care professionals in nine studies (36%), by comparing two or more groups of patients in six studies (24%), from information acquired at admission or discharge or screening in five studies (20%), and by using databases in five studies (20%).” Therefore, only around one-third of the included studies featured “surveys,” with the others being studies in which information was collected from hospital registries/databases.
As for the results, we tried to further integrate some findings. Please consider our revisions in lines 180-185 (highlighted in green).
Point 2: The discussion is also limited to a list of vaccination issues of immigrant children. The study was limited to studies conducted in Italy, which raises the question of whether a systematic review method was necessary for the study design? This point needs to be clarified in the method.
Response 2: The authors thank the reviewer for this suggestion. Our study focused on pediatric vaccinations among immigrant populations in Italy, covering a 19-year timespan. We chose a scoping review methodology to collate evidence that fits pre-specified eligibility criteria to answer our research question with the aim to minimize bias and make the available evidence accessible to decision makers in Italy. We clarified this point in the Methods section, lines 90-93 (highlighted in green).
Point 3: Second, the international sharing of digital information on vaccination histories of those from other countries would certainly be a decisive solution. But it is obvious that even if it were an ideal idea, it would be extremely difficult to achieve practically. Even in high-income countries, only a few countries have centralized management of digital information. It is too much to expect lower middle-income countries and territories, where even the management of family registers is inadequate, to manage individual vaccination histories using digital information and then share it with the rest of the world. It is necessary not only to talk about a dream, but also to clearly explain the practical challenges to realize the dream in the discussion.
Response 3: The authors agree with the reviewer regarding the ideal proposal of international sharing of digital information on vaccination histories and the practical difficulties with respect to reaching centralized management even in high-income countries. We have now included in the Discussion section comments on the practical challenges to realize this proposal. Please consider lines 346-367 (highlighted in green).
Besides, we changed the sections of the manuscript that the Editor highlighted in yellow, and we extensively reviewed the English language.
Round 2
Reviewer 3 Report
This revised manuscript appropriately corrects all of the problems identified in the previous version.